# Genetic Abnormalities of Oocyte Maturation: Mechanisms and Clinical Implications

**DOI:** 10.3390/ijms252313002

**Published:** 2024-12-03

**Authors:** Giorgio Maria Baldini, Daniele Ferri, Antonio Malvasi, Antonio Simone Laganà, Antonella Vimercati, Miriam Dellino, Domenico Baldini, Giuseppe Trojano

**Affiliations:** 1Obstetrics and Gynecology Unit, Department of Biomedical Sciences and Human Oncology, University of Bari “Aldo Moro”, 70121 Bari, Italy; antoniomalvasi@gmail.com (A.M.); antonella.vimercati@uniba.it (A.V.); miriam.dellino@uniba.it (M.D.); 2IVF Center, Momo Fertilife, 76011 Bisceglie, Italy; danieleferrimomo@gmail.com; 3Unit of Obstetrics and Gynecology “Paolo Giacone” Hospital, Department of Health Promotion, Mother and Child Care, Internal Medicine and Medical Specialities (PROMISE), University of Palermo, 90135 Palermo, Italy; antoniosimone.lagana@unipa.it; 4Department of Maternal and Child Health, Madonna delle Grazie Hospital, 75100 Matera, Italy; giuseppe.trojano@asmbasilicata.it

**Keywords:** genetic abnormalities, oocyte maturation, IVF, ICSI

## Abstract

Genetic anomalies in oocyte maturation present significant fertility and embryonic development challenges. This review explores the intricate mechanisms of nuclear and cytoplasmic maturation, emphasizing the genetic and molecular factors contributing to oocyte quality and competence. Chromosomal mutations, errors in segregation, genetic mutations in signaling pathways and meiosis-related genes, and epigenetic alterations are discussed as critical contributors to oocyte maturation defects. The role of mitochondrial defects, maternal mRNA dysregulation, and critical proteins such as NLRP14 and BMP6 are highlighted. Understanding these genetic factors is crucial for improving diagnostic approaches and therapeutic interventions in reproductive medicine, particularly for couples encountering recurrent in vitro fertilization failures. This review will explore how specific genetic mutations impact fertility treatments and reproductive success by examining the intricate oocyte maturation process. We will focus on genetic abnormalities that may disrupt the oocyte maturation pathway, discussing the underlying mechanisms involved and considering their potential clinical implications for enhancing fertility outcomes.

## 1. Introduction

Gamete maturation is essential for successful human reproduction, fertilization, embryonic development, and implantation. At the end of the egg retrieval, a series of immature oocytes is often counted, which is relatively standard. About 5–15% of the collected eggs are immature, most remaining at the germinal vesicle (GV) phase. Culturing them in vitro up to metaphase II (MII) would provide infertile patients with additional opportunities. However, the rate of GV that spontaneously reaches full maturation from stimulated cycles is still relatively low [1]. It is impossible to achieve an ideal maturation rate even by adding gonadotropins to the maturation medium [2]. Therefore, understanding the acceptable mechanisms underlying cytoplasmic and nuclear oocyte maturation is extremely important. This knowledge could improve the rate of in vitro maturation (IVM) of immature eggs, the causes of the retrieval of incompetent oocytes, or even diagnose genetic defects for which the couple would be suggested to consider oocyte donation [1]. Abnormalities in the pathway that lead the egg to the final maturation, which involves both nuclear and cytoplasmic maturation, can result in infertility and repeated unsuccessful assisted reproductive treatments (ART). Currently, the quality of oocytes or embryos is assessed only through morphological markers, which lack specific molecular markers. With the increase in IVF/ICSI patients, many cases of recurrent failure have been observed, but the genetic foundation of this phenomenon is still not well understood [2]. Recently, some mutant genes have been identified that could serve as molecular markers for oocyte/embryo quality. These anomalies mainly follow Mendelian inheritance patterns, both dominant and recessive, with some de novo mutations [3]. Genetic anomalies affecting oocyte maturation present a significant challenge in diagnosing and treating women’s infertility and embryonic development [3]. These conditions can manifest with a variety of clinical phenotypes, including polycystic ovary syndrome, premature ovarian failure, and ovarian dysfunction. Genetic factors contributing to reproductive disorders include various chromosomal abnormalities such as aneuploidy, balanced and unbalanced chromosomal rearrangements, mosaicism, copy number variations (CNVs), and pathogenic variants in multiple genes responsible for female infertility [3]. In addition to the causes mentioned above, other genetic factors, including the mitochondrial genome (mtDNA copy number and mutations), several non-coding RNAs, and epigenetic factors, could play a role in folliculogenesis, oogenesis, oocyte quality, and embryo competence [4]. However, the genetic basis of female infertility associated with meiotic defects, oocyte/embryo aneuploidies, abnormalities in oocyte development, fertilization, and early embryogenesis is not well understood [4].

This review will discuss the complex process of oocyte maturation, particularly the genetic abnormalities (Figure 1) that might affect the oocyte maturation pathway, its mechanism, and the possible clinical implications.

## 2. Genetic Abnormalities of Oocyte Maturation

### 2.1. Chromosomal Mutations

Chromosomal errors are among the most common abnormalities during oocyte maturation.

#### 2.1.1. Chromosomal Segregation Errors

Chromosomal segregation errors can lead to aneuploidy gametes, i.e., gametes with an abnormal number of chromosomes. For example, trisomy 21, which causes Down syndrome, results from an oocyte with an extra chromosome 21. An adequately assembled fusiform apparatus is critical for chromosomal segregation during cell cleavage and egg maturation, making it crucial for achieving euploidy. As a result, anomalies in the chromosomal division process, for example, during the first meiotic division in human oocytes, may result in miscarriage or failure of implantation. During meiosis in oocytes, aneuploidy could be led by the non-disjunction of entire chromosomes or sister chromatids. Nevertheless, partial or segmental aneuploidies are also significant. In addition, maternal aging is strongly correlated with an increment in meiotic chromosomal segregation mistakes, and recent studies have shown that the female reproductive lifespan is shortening due to the increasing formation of aneuploid oocytes, even in young women [5].

#### 2.1.2. Embryonic Mosaicism

In addition, chromosomal rearrangements may be involved in embryonic mosaicism, i.e., the presence of cells with different chromosomal properties within the same embryo. For example, a pool of cells may contain a chromosomal rearrangement such as a deletion or duplication, and the outcome depends on the percentage of cells with the chromosomal mutation and the content of the specific region involved in the mutation [6].

### 2.2. Genetic Mutations

Mutations in specific genes involved in oocyte maturation can impair fertility.

#### 2.2.1. Mutations in Signaling Pathways Genes

The heat shock protein 27 (HSP27) is a member of the family that may block apoptosis; it is downregulated in ovarian tissue from women with polycystic ovary syndrome (PCOS) [7]. Researchers have studied the impact of Hsp27 overexpression in vitro using oocytes from patients with PCOS by injecting an artificial green fluorescent protein (GFP) plasmid into human oocytes to increase the level of the Hsp27 protein. The results show a reduction in the maturation rate of oocytes derived from PCOS patients. Expression analysis revealed that oocyte-secreted factors (bone morphogenetic protein 15) BMP15 and growth and differentiation factor 9 (GDF9), and apoptosis-related regulators, caspases 3, 8, and 9, were significantly reduced in oocytes with Hsp27 overexpression. Hsp27 upregulation blocks oocyte maturation in PCOS patients but enhances the potential for embryonic development [7]. In addition, ovarian paracrine factors are essential in regulating egg maturation. Various paracrine factors, such as epidermal growth factor (EGF)-like ligands, leptin, brain-derived neurotrophic factor (BDNF), and glial cell line-derived neurotrophic factor (GDNF), have been demonstrated to control the resumption of meiosis in mammalian oocytes. For instance, some factors, such as BDNF and leptin, support both nuclear and cytoplasmic maturation of oocytes, thereby facilitating optimal development through to the blastocyst stage [8]. The study by Cui L. [9] showed that treatment with endothelin-1 (ET-1) led to a higher number of eggs at stage MII and observed a higher rate of regular fertilization among eggs derived from cumulus-oocyte complexes (COCs) treated with ET-1. The results show that ET-1 can promote the maturation of human eggs at the GV stage in IVM. Unusual expression of ET-1 has been documented in granulosa cells of patients with PCOS. Although these patients tend to produce more oocytes, these are frequently low quality. Therefore, understanding the role of ET-1 in oocyte maturation could lead to developing new clinical approaches for treating PCOS.

In 2023, a team of scientists published an innovative study investigating the molecular mechanisms underlying the age-related increase in oocyte aneuploidy. Using Hi C and SMART seq techniques on oocytes from young and aged mice, the researchers highlighted decreased chromosomal condensation and disruptions in meiosis-associated gene expression in metaphase I oocytes from older specimens. Further transcriptomic analysis revealed that meiotic maturation in young oocytes was accompanied by a significant upregulation of mevalonate (MVA) pathway gene expression in surrounding granulosa cells. This process appeared attenuated in aged mice. Additional experiments showed that inhibiting the mevalonate pathway in granulosa cells with statins induced severe meiotic defects and aneuploidy in oocytes from young mice. Conversely, administering the mevalonate isoprenoid geranylgeraniol mitigated meiotic defects and reduced aneuploidy in oocytes from aged mice. Mechanistically, geranylgeraniol activated luteinizing hormone (LH) receptor and epidermal growth factor (EGF) signaling in aged granulosa cells, promoting meiosis-related gene expression in oocytes. This study underscores the central role of the mevalonate pathway in granulosa cells as a critical regulator of meiotic maturation and oocyte euploidy, suggesting that age-related alterations in this pathway contribute to meiotic defects and aneuploidy [10]. Another relevant gene is PANX1 (pannexin 1), located on 11q21, composed of five exons and encoding for a protein of 422 amino acids, pannexin 1 [11]. PANX1 belongs to the family of pannexin glycoproteins involved in forming single membrane channels for molecular exchange, including adenosine triphosphate (ATP) transport. PANX1, including oocytes and early embryos, is extensively expressed in the female reproductive system. Pathogenic homozygous or compound-heterozygous variants in PANX1 result in elevated ATP release into the extracellular space, leading to oocyte degeneration. These oocytes often remain immature or degenerate shortly before or after fertilization. Heterozygous variants of PANX1 are associated with a phenotype called “oocyte death syndrome,” indicating disruption of PANX1 glycosylation and channel activity. Functional studies demonstrated that altered glycosylation and channelopathy contribute to female infertility. At the same time, heterozygous variants of PANX1 do not affect male fertility, highlighting its specific role in oogenesis and oocyte maturation [11].

#### 2.2.2. Mutations in Genes Involved in the Process of Meiosis

In oocyte maturation defect (OMD), affected patients experience primary infertility despite having regular menstrual cycles; however, they fail to produce viable oocytes, instead presenting with degenerated (atretic) or abnormal oocytes arrested at different maturation stages. Four genes have been linked to OMD: *PATL2*, *TUBB8*, *WEE2*, and *ZP1* [12]. In a study published in 2022 by Loeuillet, C. [13], oocytes from patients with *ZP1* mutations appeared shriveled and dark, suggesting that the abnormal *ZP1* protein triggered oocyte death and degeneration. Generally, patients with *ZP1* mutations exhibited degenerated or absent oocytes, unlike those with *PATL2* mutations, who displayed immature oocytes predominantly arrested at the germinal vesicle stage. Tubulin Beta class 8 (TUBB8) is one of the genes most frequently responsible for impairing oocyte maturation and embryo development. TUBB8 (tubulin, beta class 8), consisting of four exons and codes for a protein of 444 amino acids, is located on chromosome 10p15.3. TUBB8 encodes a primate-specific beta-tubulin isotype, primarily expressed during stages of human oocyte and early embryo divisions, but is absent in mature spermatozoa. Tubulin (α/β) forms the structural unit of microtubules, essential for spindle formation during meiosis [14]. Therefore, structural alterations in the TUBB8 protein are linked to oocyte maturation and early embryonic development defects. More than 130 variants of the TUBB8 gene have been identified to date. Clinical phenotypes in affected females include oocyte maturation arrest, fertilization failure, early arrest in embryonic development, and implantation failure. Specific *TUBB8* variants may lead to zygotic arrest by disrupting cell division. Heterozygous pathogenic variants in *TUBB8* commonly cause metaphase I (MI) arrest due to a dominant negative effect, though early embryonic arrest has also been reported. Homozygous or compound heterozygous variants in *TUBB8* are associated with milder phenotypes, such as fertilization failure (MII arrest), arrest in early embryonic development, and failure in zygotic cleavage. Functional studies have shown that pathogenic *TUBB8* variants disrupt the chaperone-mediated folding and assembly of the α/β-tubulin heterodimer, impairing microtubule structure and causing defects in spindle assembly, ultimately leading to arrested oocyte maturation [15]. Around 30% of maturation arrests are caused by mutations in TUBB8, indicating the dominant role of this gene in the disease. The human β-tubulin family includes nine β-tubulin isotypes, with *TUBB8* being the only one specifically expressed in human oocytes and early embryos, indicating its role in spindle assembly and likely contributing to the unique aspects of oocyte maturation in primates [14]. To date, variations of the genetic sequence of the gene *TUBB8* have been associated with five distinct phenotypes: the arrest of oocyte maturation, PB1 oocytes that fail to be fertilized, PB1 oocytes that can be fertilized but the embryos fail to divide, PB1 oocytes that can be fertilized and the embryos can divide but with the subsequent arrest of embryonic development, and some implantable embryos can be obtained but fail to lead to pregnancy after implantation [14,15]. In the study by Yao, Z. [14], a total of five heterozygous/homozygous mutations in TUBB8 were found in eleven infertile women (p.A313V, p.C239W, p.R251Q, p.P358L, and p.G96R). Wild-type (WT) genetic constructs and four mutants were transfected into Hela cells. Immunofluorescence assays demonstrated that HeLa cells transfected with the p.C239W, p.R251Q, or p.G96R mutations exhibited disrupted microtubule structures, indicating a marked alteration in microtubule network organization compared to wild-type (WT) cells. In this study, TUBB8 gene variants were identified in 31.96% (109 out of 341) of participants who experienced arrested oocyte maturation. Additional mutations in TUBB8 were subsequently detected, which caused the oocytes or embryo to stop. While some of these women may experience biochemical or ectopic pregnancies, no live births or ongoing pregnancies have been reported to date [14]. Given the expanding range of TUBB8 mutations affecting human oocyte development, fertilization, and early embryonic development, screening for TUBB8 mutations holds significant value for evaluating PB1 oocyte function and offering precise diagnoses for infertile patients experiencing recurrent IVF/ICSI failure [13].

The Thyroid Hormone Receptor Interactor 13 (TRIP13) gene is located on the short arm of chromosome 5 (5p15.33). It consists of 13 exons that encode a 432-amino-acid protein, AAA-ATPase, a vital spindle assembly checkpoint (SAC) component. *TRIP13* is widely expressed in various tissues, including germ cells, and is crucial in mitosis and meiosis. During mitosis, the TRIP13 protein aids in spindle checkpoint silencing, while in meiosis, it is involved in recombination pathways. Specific recessive pathogenic variants of *TRIP13* that impair mitosis and meiosis have been linked to distinct diseases, including Wilms tumors and female infertility [14]. Homozygous nonsense variants or pathogenic mutations that disrupt splicing lead to a complete loss of TRIP13 function, resulting in abnormal mitosis. In contrast, pathogenic missense variants primarily impact oocyte meiosis, causing a slight reduction in protein levels. Homozygous and compound heterozygous missense variants in *TRIP13* are associated with female infertility, characterized by oocyte meiotic arrest and abnormal zygotic cleavage. Notably, injecting *TRIP13*-encoding RNA into the oocytes of affected females has been shown to restore function, presenting a potential therapeutic avenue [16].

The Topoisomerase II-Associated Protein (PATL2) gene, located on the long arm of chromosome 15 (15q21.1), comprises 15 exons and encodes a highly conserved, oocyte-specific mRNP repressor protein that regulates the expression of genes critical for oocyte maturation and early embryonic development. *PATL2* expression is specifically high in immature oocytes, decreasing significantly as the oocyte matures. Homozygous or compound heterozygous variants in PATL2 lead to infertility due to oocyte arrest at the germinal vesicle (GV) or metaphase I (MI) stage, fertilization failure, and early embryonic developmental arrest. In a study by Liu, R. [17], PATL2 mutations were investigated in 40 patients with infertility due to oocyte maturation arrest. Genomic DNA was extracted from peripheral blood, and whole-exome sequencing was performed, with PATL2 mutations subsequently confirmed by Sanger sequencing. These mutations were associated with ovarian phenotypes, including GV arrest, MI arrest, and morphological abnormalities [17]. TBPL2, a general transcription factor specific to vertebrate oocytes, plays an essential role in oocyte development. Transcriptome sequencing of affected oocytes revealed a marked downregulation of genes critical for oocyte maturation and fertilization, suggesting that mutations in TBPL2 lead to widespread gene expression changes in oocytes. Findings have identified a homozygous splicing mutation in TBPL2 potentially linked to defects in human oocyte maturation [18]. In a 2019 study, researchers identified a rare, pathogenic homozygous missense mutation (c.895T>C; p.C299R) in TBPL2 in two infertile sisters from a consanguineous family, both experiencing oocyte maturation arrest and degeneration. Whole-exome sequencing confirmed that this mutation impairs the transcription initiation function of TBPL2, and it is associated with female infertility characterized by oocyte maturation arrest and degeneration [19]. The Cell Division Cycle 20 (CDC20) gene, located on the short arm of chromosome 1 (1p34.2), contains 11 exons and encodes a 499 amino-acid protein expressed in many tissues. CDC20 functions as a co-activator of the anaphase-promoting complex/cyclosome (APC/C), essential for transitioning from metaphase to anaphase during mitosis and meiosis. By binding to APC/C, CDC20 forms the APC/C^CDC20 complex, which promotes the degradation of securin and cyclin B, inactivation of cyclin-dependent kinases CDK1 and CDK2, and separation of sister chromatids, thereby facilitating the metaphase-anaphase transition. CDC20 expression peaks during the G2/M phase of the cell cycle. Containing seven WD40 repeats at its C-terminus, CDC20 is crucial for protein-protein interactions necessary for mitotic and meiotic progression. Recent research has linked biallelic mutations in CDC20 to female infertility, characterized by impaired oocyte maturation and embryonic development. Specifically, the nonsense mutation p.R262 leads to either absent protein production or a truncated protein missing five of the C-terminal WD40 repeats. In contrast, the p.A211T mutation likely disrupts the formation of a deep hydrophobic pocket, impairing CDC20’s binding to APC/C substrates [20]. The anaphase-promoting complex or cyclosome (APC/C) is essential for transitioning from metaphase to anaphase. CDC23 (cell division cycle 23) is one of the critical subunits of APC/C. In vivo studies using Cdc23^Y329C/Y329C mice successfully replicated the phenotype observed in patients, showing reduced expression of CDC23 and APC4 and an accumulation of securin and cyclin B1 in oocytes. Treatment with AZ3146 was effective in rescuing this phenotype [21]. Data from IVF patients at Istanbul Memorial Hospital between 2015 and 2021 were analyzed using statistical methods to identify infertile endophenotypes, such as low egg maturation rates, low fertilization rates, and preimplantation development arrest. This dataset included 11,221 couples [22]. Following this analysis, 28 infertile women diagnosed with OOMD/PREMBL were selected for whole-exome sequencing (WES) on their genomic DNA. Seven of these 28 women (25%) were found to be homozygous carriers of pathogenic missense variants in known candidate genes for OOMD/PREMBL, including PATL2, NLRP5 (*n* = 2), TLE6, PADI6, TUBB8, and TRIP13. New gene-disease associations were identified, including a woman with a low oocyte maturation rate who was found to be a homozygous carrier of high-impact variants in ENSA, a gene essential for the prophase I meiotic transition in mice [22]. In a study conducted by Huang, L. [23], the genetic causes of primary infertility were investigated in 12 women primarily characterized by oocyte maturation abnormalities and subsequent early embryonic arrest. A novel homozygous frameshift variant (p. V429Efs30) in NLRP5 and compound heterozygous variants, including a new frameshift variant (p. A297Efs20) and a recurrent variant (c.223-14_223-2delCCCTCCTGTTCCA) in PATL2, were identified in two unrelated affected individuals. Quantitative RT-PCR demonstrated a significant decrease in mutant NLRP5 mRNA levels. Additionally, truncated proteins resulting from these mutations in NLRP5 and PATL2 were predicted to be non-functional due to the loss of most or all critical functional domain regions. Notably, only 1% of patients carrying biallelic pathogenic variants in the NLRP7 gene were able to achieve live births through spontaneous conception, with the majority requiring IVF/ICSI procedures using donor eggs [24]. Zhao L. [25] identified LHX8 as a novel mutant gene responsible for halting the maturation of human oocytes. Most variants of TUBB8 were found to be missense pathogenic with dominant-negative effects, while homozygous or compound heterozygous variants in PATL2 and TRIP13 produced defective proteins. In the case of LHX8, loss-of-function (LOF) variants resulted in complete functional loss of an allele, demonstrating a distinct causal effect known as haploinsufficiency. This mechanism is associated with female infertility characterized by oocyte maturation arrest. Other ovarian genes may also interact with LHX8, suggesting further investigation into additional genomic variants is warranted. Future studies could involve constructing corresponding variants in mouse models for functional assays, enhancing our understanding of the interplay between genetic factors and reproductive health. In the context of female infertility, a study by Hamdan M. [26] in 2019 demonstrated that mouse oocytes arrest in meiosis I due to DNA damage, which activates DNA damage response (DDR) pathways and the spindle assembly checkpoint (SAC). The researchers utilized follicular fluid from patients with endometriosis, characterized by elevated levels of reactive oxygen species (ROS), to induce DNA damage and activate the DDR–SAC pathway. Notably, only follicular fluid from endometriosis patients—not from controls—produced ROS and caused DNA damage in the oocytes [26]. This activation of the ATM kinase led to the arrest of SAC-mediated metaphase I. Significantly, the completion of meiosis I could be restored using ROS scavengers, highlighting ROS as a primary trigger for oocyte arrest and suggesting a novel clinical treatment approach. This study underscores the clinical relevance of DDR-induced SAC in oocytes, providing valuable insights into how oocytes respond to prevalent human diseases, such as endometriosis, and the associated reductions in fertility. The RAE2 gene (WEE1 homolog 2) on chromosome 7q34 consists of 12 exons and encodes a 567-amino-acid protein. WEE2 belongs to the WEE family of protein kinases and is specifically expressed in oocytes. This tyrosine kinase plays a crucial role in regulating meiosis during prophase I (germinal vesicle [GV] stage) and metaphase II (MII). WEE2 maintains meiotic arrest in the GV phase by inhibiting the maturation-promoting factor (MPF) through the inactivation of CDK1. During MII, WEE2 facilitates the exit from MII [27]. Reduced expression of WEE2 in both human and murine oocytes has been associated with elevated MPF activity, impairing MII outcomes and, consequently, fertilization potential. Homozygous or compound heterozygous variants in WEE2 have been linked to female infertility due to fertilization failure. Although the oocytes from these patients may appear morphologically normal, they fail to form a zygote after intracytoplasmic sperm injection (ICSI) due to impaired phosphorylation of WEE2 and CDK1, leading to implantation block (IBD) termination. Functional studies indicate that pathogenic variants of WEE2 result in decreased WEE2 levels, compromising proper oocyte development and fertilization [27]. The Rac1 gene, a member of the Rho GTPase family, has been shown to regulate polarity and asymmetric division in mouse oocytes in vitro. In studies utilizing conditional gene knockout (CKO) technology, deletion of the Rac1 gene in mouse oocytes disrupted oocyte maturation, affecting both polarity establishment and asymmetric division; however, the mutant mice exhibited normal fertility [28]. In humans, oocytes typically remain arrested in meiotic prophase I until puberty. During the menstrual cycle, an increase in luteinizing hormone (LH) raises phosphodiesterase 3 (PDE3) levels in oocytes, which hydrolyzes cyclic adenosine monophosphate (cAMP) and simultaneously activates cell division cycle gene 25 (Cdc25). This process leads to the activation of maturation-promoting factors (MPFs), a dimer composed of cyclin-dependent kinase 1 (CDK1) and cyclin B [29]. Consequently, meiosis resumes in the oocytes, leading to the germinal vesicle (GV) breakdown. If mutations or deletions occur in Cdc25, PDE3, or CDK1, the oocyte fails to resume meiosis and remains arrested in the GV phase. The anaphase-promoting complex/cyclosome (APC/C) serves as a ubiquitin ligase that releases separase to facilitate the separation of homologous chromosomes. The spindle assembly checkpoint (SAC) signaling pathway regulates chromosome alignment and segregation. Disruption of normal APC/C and SAC regulation can hinder the transition from the metaphase I (MI) phase to anaphase and telophase. When molecules (APC/C and SAC) within these pathways, which belong to families of proteins with high sequence homology, encounter issues, other proteins may compensate for their function to varying extents. This inconsistent compensation can result in oocytes from certain patients being arrested at different stages, leading to a mixed arrest phenotype [29]. The REC114 gene (meiotic recombination protein 114), located on chromosome 15q24, consists of six exons and encodes a 266-amino-acid protein essential for initiating meiotic chromosomal recombination through the formation of double-stranded breaks (DSBs). SB formation is critical for chromosomal segregation during meiosis I and is regulated by several genes, including IHO1, FROM4, and REC114 [30]. REC114 forms a stable complex with MEI4 and IHO1 from the onset of prophase I until the synapsis at the pachytene stage in models such as Saccharomyces cerevisiae. Homozygound-heterozygous variants in REC114 have been associated with the arrest of oocyte maturation and early embryonic developmental arrest. Moreover, homozygous pathogenic variants affecting REC114 splicing have been linked to recurrent hydatidiform mole formation. These findings underscore the critical role of REC114 in meiosis and early embryonic development, significantly influencing female fertility outcomes [30]. In their study, Wang W. [31,32] aimed to identify the genetic causes of oocyte maturation arrest (OMA). They focused on PABPC1L (cytoplasmic poly(A)-binding protein 1-like), which encodes a protein that binds to elongated poly(A) tails to stabilize polyadenylated mRNAs. Female mice deficient in Pabpc1l are sterile; their oocytes are dysmorphic and fail to complete maturation due to altered translational activation of maternal mRNAs. The authors conducted a mouse zygote transcriptome analysis to elucidate the mechanisms underlying maturation arrest in PABPC1L-deficient oocytes. This analysis revealed an upregulation-MAPK pathway crucial for inducing oocyte MII arrest. The authors demonstrated that microinjection of human wild-type (WT) MOS cRNA into WT mouse zygotes replicated the knockout (KI) phenotype [31]. Confirmation of genetic causality and studies involving the overexpression of MOS in PABPC1L variant oocytes suggest potential therapeutic approaches. Early oocytes or embryos from women carrying pathogenic variants could be “rescued” by introducing Mos siRNA or MAPK inhibitors. An alternative approach could involve modeling previous studies in mice, where microinjection of Pabpc1l mRNA into Pabpc1l-deficient oocytes encased in the preantral follicle successfully saved oocyte maturation. However, this method did not yield positive results with denuded GV oocytes.

### 2.3. Epigenetic Alterations

#### 2.3.1. DNA Methylation

Epigenetic modifications play a pivotal role in regulating gene expression independently of alterations to the DNA sequence (Figure 2), and these modifications are integral to the maturation of oocytes. DNA methylation represents a primary epigenetic mechanism wherein shifts in methylation patterns can disrupt the gene expression necessary for proper oocyte development [33]. Similarly, histone modifications are essential, as changes in post-translational modifications of histones can alter chromatin architecture and DNA accessibility, thereby impacting processes fundamental to oocyte maturation [34]. The maternal-to-zygotic transition (MZT) marks a critical developmental phase where control of embryonic development shifts entirely from maternal to zygotic genome activation. This transition is intrinsically linked to chromatin remodeling and specific histone modifications. Among these, H2AK119ub1 plays a significant regulatory role in determining chromatin configuration and function [34].

#### 2.3.2. Histone Changes

In 2022, Rong Y. conducted a comprehensive study on the distribution dynamics of H2AK119ub1 throughout the maternal-to-zygotic transition (MZT) in mice, observing that H2AK119ub1 accumulates extensively in fully matured oocytes, particularly at transcription start sites (TSS) of maternal genes (Figure 2). Following meiotic resumption, however, H2AK119ub1 levels diminish significantly at the genomic level [35]. Genetic analyses identified ubiquitin-specific peptidase 16 (USP16) as the primary de-ubiquitinase responsible for modulating H2AK119ub1 levels in mouse oocytes [34]. While conditional USP16 knockout in oocytes does not affect their survival, growth, or progression through meiosis, its absence impedes zygotic genome activation and reduces developmental competence after fertilization [35]. This effect likely results from excessive deposition of maternal H2AK119ub1 on the zygotic genome. Thus, the USP16-dependent reduction in H2AK119ub1 levels during oocyte maturation is critical for proper zygotic genome reprogramming and for activating the transcription of essential early zygotic genes [35]. Long non-coding RNAs (lncRNAs), known to participate in diverse biological processes—including regulation of gene expression, chromatin organization, mRNA transport, and modulation of protein activity—are actively expressed in both oocytes and early embryos. Their expression patterns shift dynamically following the activation of the embryonic genome during the early stages of human embryonic development [36]. Previous studies have demonstrated that lncRNAs contribute to critical aspects of early embryogenesis, such as the induction and maintenance of pluripotency, X chromosome inactivation, and gene imprinting. Notably, the expression levels of certain lncRNAs in cumulus cells correlate with oocyte and early embryo quality, suggesting their potential as non-invasive biomarkers for predicting oocyte developmental potential [36]. In a recent study, lncRNA sequencing was conducted on RNA pools from 20 oocytes per group—recurrent arrest of oocyte maturation (ROMA), germinal vesicle (GV), metaphase I (MI), and metaphase II (MII)—using bioinformatics software to assess differential lncRNA expression between normal and ROMA oocytes. This analysis revealed 17 downregulated and three up-regulated lncRNAs in ROMA oocytes, with co-expression analysis indicating notable downregulation of NEAT1 and NORAD, which belong to the same family of RNAs [36]. In their study, Wyse B.A. examined the functional distinctions in cumulus cells (CCs) surrounding oocytes at different maturation stages, investigating their potential as non-invasive biomarkers for oocyte quality assessment [37]. CCs from 18 patients with oocytes at varying maturation stages within the same cycle were analyzed. The findings revealed that CCs surrounding mature oocytes exhibited greater transcriptional synchronization than those associated with immature oocytes. Furthermore, CCs from mature oocytes showed reduced transcriptional activity, with fewer transcripts detected than CCs from immature oocytes [37]. These transcriptional differences involved numerous biological processes, including cell cycle regulation, steroid metabolism, apoptosis, extracellular matrix remodeling, and inflammatory pathways. This suggests that the transcriptional profile of CCs may reflect the maturation state and quality of the associated oocyte, offering valuable insights for reproductive assessment [37]. Cumulus cells and the remodeling of their extracellular matrix (ECM) surrounding oocytes are vital for the maturation and ovulation of oocytes in the ovary. A critical component of this process is the extracellular metalloprotease ADAMTS1, along with its partner VERSICAN, which is essential for the structural remodeling of the cumulus-oocyte complex (COC) [38]. Notably, the expression of TIMP3, an inhibitor of metalloproteinases, is significantly reduced in cumulus cells during in vitro maturation. When TIMP3 was suppressed using specific small interfering RNAs, its expression decreased, increasing levels of ADAMTS1 and VERSICAN [38]. Additionally, MiR-21, a microRNA significantly elevated in cumulus cells throughout COC maturation, plays a regulatory role in this process. A decrease in MiR-21 during COC maturation led to an upregulation of TIMP3 and a corresponding reduction in ADAMTS1 and VERSICAN, linked to diminished cumulus cell expansion and a lower proportion of oocytes reaching the MII stage. In contrast, the overexpression of MiR-21 resulted in decreased TIMP3 levels and increased ADAMTS1, thereby enhancing cumulus cell expansion and promoting oocyte maturation. Thus, MiR-21 exerts a crucial influence on cumulus expansion and oocyte maturation by inhibiting TIMP3, which subsequently elevates ADAMTS1 and VERSICAN during the in vitro maturation of COCs [38]. In a study examining the expression levels of miR-514 and miR-642b, along with their respective candidate target genes, EGFR and PTGER2, in cumulus cells (CCs) from immature and mature oocytes in patients with polycystic ovary syndrome (PCOS), quantitative real-time PCR was employed. The expression levels of these miRNAs and their target genes were compared between CCs from germinal vesicle (GV) oocytes and those from metaphase II (MII) oocytes [36]. The analysis revealed that CCs surrounding GV oocytes exhibited higher mRNA levels of both EGFR and PTGER2 while displaying lower expression levels of miR-514 and miR-642b than CCs surrounding MII oocytes. Consequently, assessing these miRNAs in CCs presents a promising biomarker for predicting oocyte competence in patients with PCOS [39].

### 2.4. Other Factors

#### 2.4.1. Mitochondrial Defects

Mitochondria play a vital role in energy production, and any defects in their function can adversely affect the maturation and quality of oocytes (Figure 3). In older women, oogenesis—the process of egg cell development—experiences a prolonged arrest during prophase I, leading to the accumulation of free radicals associated with aging within the human genome [40]. The inherent instability of the mitochondrial genome, combined with prolonged exposure to reactive oxygen species, renders mitochondrial DNA (mtDNA) vulnerable to various mutations. Consequently, numerous germ cells possess mtDNA mutations, with common alterations such as a 4.9 kb deletion being particularly prevalent. These mutations result in increased mtDNA copies as a compensatory mechanism; however, the proportion of mutant mtDNA is typically low relative to the total mtDNA content in mature oocytes, indicating low heteroplasmy. It has been postulated that even minimal mtDNA mutations and aneuploidies can lead to elevated mtDNA copy numbers in oocytes as an adaptive response. Deficiencies in ATP production can impair meiotic spindle formation and negatively impact chromosomal segregation, often increasing aneuploidies that can lead to the death of oocytes or embryos. This observation suggests a purifying selection process that eliminates oocytes harboring mtDNA mutations. Most harmful variants of the mitochondrial genome are negatively selected during oogenesis, indicating a germline bottleneck in their transmission to the next generation [41]. The prevalence of mitochondrial DNA (mtDNA) variants in the MT-ND1 gene and the D-loop region observed in oocytes from patients with recurrent miscarriages indicates that mtDNA mutations may represent one of the genetic factors hindering embryonic development [42]. In contrast to the robust purifying selection that acts against deleterious mutations in protein-coding regions of mtDNA, the adverse selection against specific variants of tRNA and rRNA genes is less pronounced. This diminished selection may be attributed to their mild pathogenic effects and/or lower heteroplasmy levels. Consequently, the quantity of mtDNA copies and the presence of specific gene variants in oocytes and follicular cells may be associated with oocyte quality and the incidence of aneuploidies. These factors significantly impact the embryo’s capacity for further development and, consequently, affect pregnancy progression. Women exhibiting a reduced quantity and/or lower quality of mtDNA in their gametes tend to have diminished ovarian reserves and decreased fertility. This condition serves as a predictor of poor embryo development and implantation potential, ultimately increasing the risk of early miscarriage [42]. Although the precise mechanisms underlying this phenomenon remain inadequately understood, it is increasingly evident that both qualitative and quantitative changes in the mitochondrial genome may play a crucial role in ovarian aging and reduced fertility [42]. Several factors must be considered when evaluating mitochondrial DNA (mtDNA) integrity, including damage caused by oxidative stress, the accumulation of acquired mtDNA mutations, the influence of inherited mtDNA mutations, and modifications in the mitochondrial stress response mechanisms. Recently, mtDNA content within embryonic cells has gained attention as a potential biomarker for assessing embryo implantation potential [43]. Filali M. [44] investigated the influence of BAX and BCL2 expression in cumulus cells on the developmental competence of in vitro matured oocytes. BAX (BCL2-associated protein X) and BCL2 (B-cell leukemia/lymphoma gene) are members of the BCL2 gene family, functioning as pro-apoptotic and anti-apoptotic proteins, respectively, and they are integral to the mitochondrial-dependent apoptosis pathway. The study involved the analysis of 100 cumulus-oocyte complexes (COCs) harvested from unprimed ovaries of 13 women diagnosed with polycystic ovary syndrome (PCOS) who were undergoing in vitro maturation (IVM). The oocyte was separated from the cumulus cells after maturing the COCs in a specialized medium for 24 h. The mRNA levels of BAX and BCL2 were quantified in each COC using a real-time polymerase chain reaction. The results demonstrated a significantly higher expression of BCL2 mRNA in cumulus cells associated with mature oocytes than those linked to immature oocytes, while the BAX mRNA levels remained stable. Furthermore, increased BCL2 mRNA content was noted in cumulus cells surrounding fertilized oocytes. These findings indicate a strong correlation between BCL2 expression and the capacity of oocytes to achieve nuclear maturation and successfully undergo fertilization [44].

#### 2.4.2. Maternal mRNA

Exome sequencing has been employed to uncover the genetic factors contributing to defects in oocyte maturation (Figure 3). For instance, a homozygous variant c.853_861del (p.285_287del) in the ZFP36L2 gene was identified in an individual from a consanguineous family exhibiting oocyte maturation defects. ZFP36L2 encodes an RNA-binding protein critical in regulating maternal mRNA decay and facilitating oocyte maturation. In vitro studies revealed that this variant reduced ZFP36L2 protein levels in oocytes due to mRNA instability, which may impair its function in degrading maternal mRNAs. Previous research has established that pathogenic variants in ZFP36L2 are linked to early embryonic arrest. Identifying this novel ZFP36L2 variant indicates its potential as a genetic diagnostic marker for individuals experiencing oocyte maturation defects [45]. Fully developed oocytes exhibit transcriptional quiescence; however, they synthesize and store significant quantities of maternal mRNAs throughout their growth phase. Recent investigations have uncovered that these maternal mRNAs are organized within a structure called the mitochondria-associated ribonucleoprotein domain (MARD). Nonetheless, the specific components and functions of MARD remain largely ambiguous [46]. Research demonstrates that the knockout of LSM14B hinders the proper storage and timely degradation of several mRNAs, including cyclin B1, Btg4, and others typically activated during meiotic maturation. This disruption mainly affects the assembly of MARD during both oocyte growth and meiotic maturation. Consequently, oocytes lacking LSM14B exhibit diminished mRNA storage and clearance levels, preventing their advancement into meiosis II and ultimately resulting in female infertility. These findings underscore the crucial role of LSM14B in the formation of MARD and establish a connection between this structure, mRNA degradation, and oocyte meiotic maturation [46]. BTG4 (B cell translocation gene 4), situated on the long arm of chromosome 11 at position 11q23.1, comprises five exons and encodes a protein consisting of 223 amino acids. This gene is primarily expressed in oocytes, ovaries, and early-stage embryos. The BTG4 protein is known to interact with the CCR4-NOT complex, which facilitates the degradation of maternal mRNAs and expedites the maternal-zygotic transition (MZT) by promoting mRNA deadenylation [47]. During mammalian oogenesis, maternal mRNAs are synthesized and accumulated to support the meiotic maturation of oocytes. These mRNAs undergo degradation during the meiotic recovery phase, with approximately 90% eliminated throughout the MZT process [48]. Following mRNA degradation, the zygotic genome becomes activated, marking the transition from the zygote formation stage to that of a two-cell embryo [47]. The principal mechanism for mRNA degradation involves shortening the poly(A) tail, a process referred to as deadenylation. A pivotal enzyme complex in this pathway is the CCR4-NOT deadenylase, which does not directly bind to mRNA. Instead, it is recruited to target mRNAs through adapter proteins from the BTG/TOB family, including BTG4. Dysfunction of BTG4 results in a marked delay in the degradation of maternal mRNAs during the maternal-zygotic transition (MZT). Pathogenic variants within the BTG4 gene disrupt its interaction with the CCR4-NOT complex, leading to the arrest of the first zygotic cleavage due to the accumulation of maternal mRNAs. Female Btg4-null mice generate morphologically normal oocytes; however, these oocytes are arrested at the single-cell stage, resulting in sterility, while their male counterparts exhibit normal fertility. Zheng W. identified homozygous pathogenic variants in the BTG4 gene associated with the ZCF phenotype in four unrelated female patients [48].

#### 2.4.3. Others Important Proteins

NLRP14 (NLR family pyrin domain containing 14) is a protein part of the NOD-like receptor (NLR) family, which is critical in regulating innate immune responses and inflammatory mechanisms. Specifically, NLRP14 is predominantly expressed in mammalian oocytes and early-stage embryos, indicating a distinct function in embryonic reproduction and development [49]. The maternal absence of NLRP14 results in infertility characterized by defects in oocyte maturation and early embryonic arrest (EEA). Eliminating NLRP14 adversely affects oocyte competence due to diminished cytoplasmic and nuclear maturation. Notably, NLRP14 sustains the cytoplasmic levels of UHRF1 by safeguarding it from proteasome-mediated degradation and stabilizing it to prevent its translocation to the nucleus within the oocyte. Bone morphogenetic protein 6 (BMP6) is a regulatory peptide produced by oocytes and granulosa cells, playing a vital role in the local regulation of folliculogenesis and follicular maturation [50]. An analysis utilizing reverse transcription polymerase chain reaction (RT-PCR) conducted on 354 single cumulus cell (CC) samples from 48 women examined the correlation between BMP6 mRNA expression in CCs and the developmental potential of oocytes. The findings revealed that the BMP6 protein is predominantly localized in the oocytes of preantral follicles and the granulosa cells of antral follicles. Notably, BMP6 mRNA expression was significantly increased in oocytes compared to CCs and mural granulosa cells (mGCs) of preovulatory follicles (*p* < 0.01), with BMP6 protein levels being higher in CCs than in mGCs (*p* < 0.05). Additionally, BMP6 mRNA expression was greater in CCs associated with immature oocytes than those linked to mature oocytes (*p* < 0.05). However, no significant association was identified between BMP6 mRNA expression in CCs and oocyte fertilization, embryo morphological assessment, or implantation outcomes. In summary, BMP6 is predominantly expressed in oocytes throughout all stages of human follicular development, and the expression of BMP6 mRNA in CCs may exhibit a negative correlation with oocyte maturation. Consequently, BMP6 expression can be a biomarker for assessing oocyte maturation [50]. The zona pellucida (ZP) constitutes a vital glycoprotein matrix that plays significant roles in oocyte maturation, ovulation, and fertilization, including inducing the acrosomal reaction, preventing polyspermia, and supporting the early stages of embryonic development [47]. In humans, the ZP resembles that of rats and comprises four glycoproteins: ZP1, ZP2, ZP3, and ZP4, each encoded by specific genes. These ZP genes are exclusively expressed during oocyte development throughout follicular growth. The activation of ZP genes during this developmental phase leads to the synthesis of ZP glycoproteins, which assemble to create the zona pellucida. ZP2-ZP3 dimers polymerize into long fibrils cross-linked by ZP1, forming a robust extracellular matrix structure. Pathogenic variants in the human ZP1, ZP2, and ZP3 genes can result in the thinning or absence of the ZP, empty follicle syndrome, or abnormal oocyte formation, thereby contributing to female infertility [51]. This highlights the essential function of normal ZP1 protein in forming the ZP and natural fertilization processes. In cases of certain IVF/ICSI failures accompanied by the absence of ZP, heterozygous or compound-heterozygous variants have been identified in the ZP1 gene. Conversely, heterozygous variants in ZP4 do not seem to affect infertility linked to ZP absence. Functional studies demonstrate that biallelic pathogenic variants disrupt the assembly of ZP proteins, leading to oocyte degeneration. Interestingly, while some investigations suggest no correlation between pathogenic variants in ZP4 and infertility, ZP morphology in Zp4-/- rats remains intact. Consequently, current evidence is inadequate to clarify the direct involvement of the ZP4 gene in disorders related to ZP formation [52]. Maintaining the integrity of chromatin remodeling is particularly crucial in oocytes due to their heightened vulnerability to aneuploidy, especially during the first meiotic division. The investigation conducted by Swain [53] emphasizes that Aurora kinases (AURKs), a family of serine/threonine kinases, play a significant role in regulating various structural components and mechanical processes associated with dynamic chromatin remodeling. Consequently, analyzing AURKs during oocyte maturation is valuable for identifying the underlying factors and molecular signaling pathways contributing to atypical oocyte chromosome modifications. When AURKs were inhibited during the transition from prophase to metaphase I (MI) using the ZM447439 inhibitor (ZM), there was no impact on the rupture of the germ vesicle. However, the resulting meiotic spindles exhibited malformation, and both the microtubule organizing centers and chromatin appeared disorganized [53]. Chromosomal assessments of MI oocytes revealed that the inhibition of AURK led to abnormal chromosomal condensation. Furthermore, inhibiting AURK during the transition from prophase I to metaphase II (MII) hindered the completion of MII, prevented polar body extrusion, and induced aberrant chromatin remodeling.

## 3. Materials and Methods

In this review, we identified studies that describe or assess the genetic pathways of oocytes and their effects on female fertility. We searched PubMed, Scopus, ResearchGate, Web of Science, and preprint archives for research articles published up to July 2024 using the following terms: “Genetic abnormalities in oocyte maturation”, “Chromosomal mutations and oocyte maturation”, “errors in segregation and oocyte maturation”, “genetic mutations and oocyte maturation”, “epigenetic alterations and oocyte maturation”, and “genetic oocyte maturation defect”. Four authors independently reviewed research titles to eliminate studies that did not meet the issue before reviewing selected studies’ full abstracts and texts.

## 4. Conclusions

Advanced genetic diagnostic techniques, such as DNA sequencing and pre-implantation genetic testing (PGT-A), can identify these abnormalities in oocytes or embryos. Understanding these abnormalities and the list of genes (Table 1) involved is crucial for fertility treatment and reproductive medicine decisions, such as selecting oocytes and embryos for use in in vitro fertilization (IVF) cycles. Genetic abnormalities in oocyte maturation can result from various errors and genetic mutations. These defects can have profound implications for fertility and embryonic development [54], making it essential to understand and diagnose them early for effective treatment of fertility problems.

## Figures and Tables

**Figure 1 ijms-25-13002-f001:**
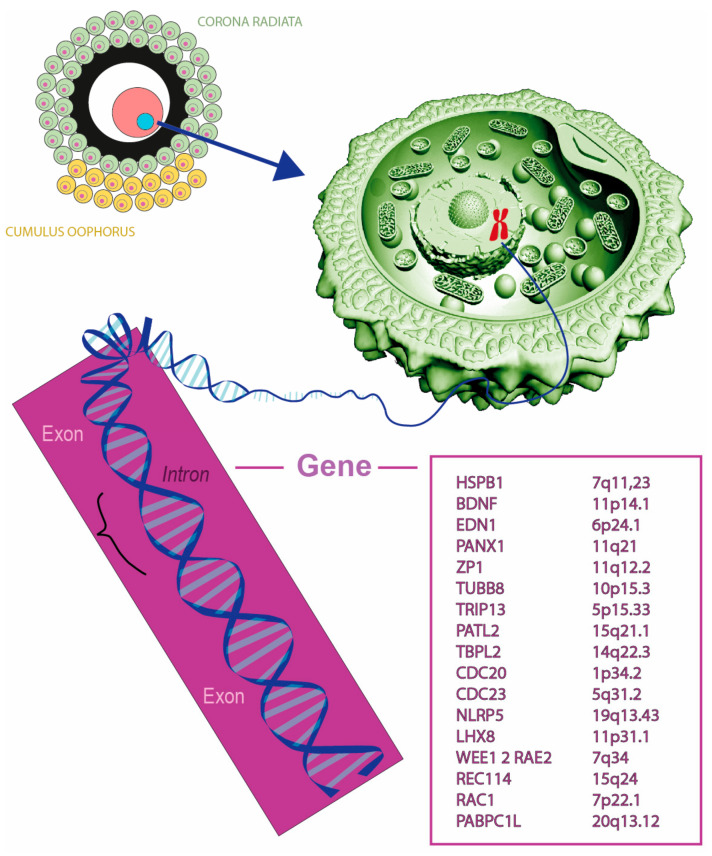
Illustration showing the structure of an oocyte surrounded by corona radiata and cumulus oophorus cells, highlighting critical genes involved in oocyte maturation and fertility, with their chromosomal locations.

**Figure 2 ijms-25-13002-f002:**
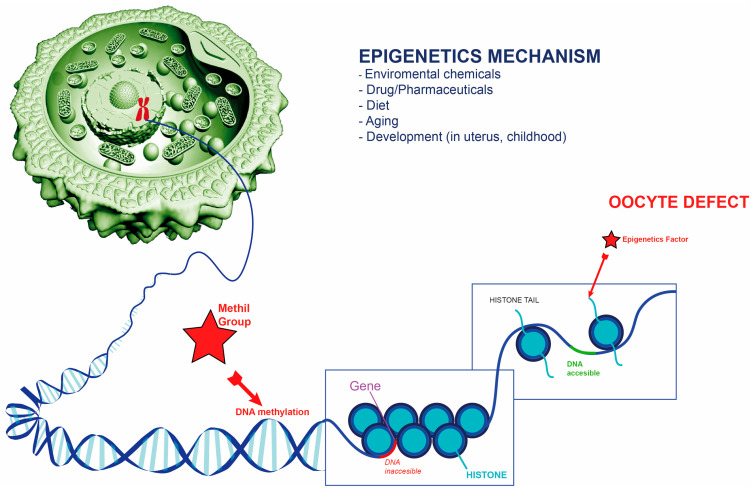
Epigenetics mechanism in oocyte defect. Mechanism of methylation through DNA alterations and epigenetic damage via histones on accessible genes.

**Figure 3 ijms-25-13002-f003:**
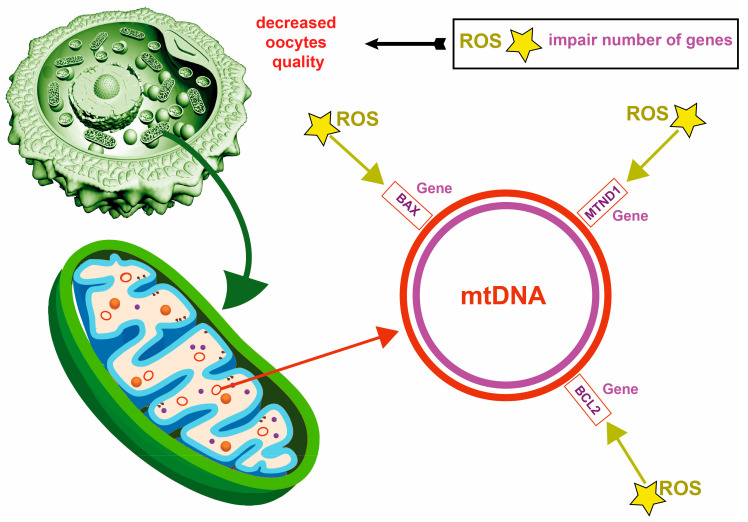
Mitochondrial and mtDNA defect, decrease in oocyte quality, through alterations of some genes by ROS.

**Table 1 ijms-25-13002-t001:** List of genes involved in the anomalies of the oocyte maturation [55].

GENE	POSITION	Exons	Encoded Protein	N. Aminoacid	Alteration Fertility	Related Pathologies
*HSPB1*	7q11.23		Heat Shock Protein Hsp27		Oocyte maturation inhibition	Distal Motor Neuropathy and Charchot Marie Tooth Disease
*BDNF*	11p14.1		BDNF brain-derived neurotrophic factor	247	Nuclear and cytoplasmic maturation of oocytes	Obsessive-compulsive disorder, schizophrenia, dementia etc
*EDN1*	6p24.1		ET-1 Endothelin 1	21	Maturation abnormalities Polycystic ovary syndrome	Auricle condylar syndrome
*PANX1*	11q21	5	Pannexina 1	422	Oocyte death syndrome Female infertility	-
*ZP1*	11q12.2		ZP1 zona pellucida protein		Death and oocyte degeneration Withered and dark oocytes Sindrome empty follicles	-
*TUBB8*	10p15.3	4	Tubulin beta 8	444	Defects in oocyte maturation Early embryonic arrest	-
*TRIP13*	5p15.33	13	AAA-ATPasi	432	Regulates mitosis and meiosis Female infertility	Male Infertility
*PATL2*	15q21.1	15	Topoisomerase II associated protein	543	Arrest of oocyte maturation at the GV or MI stage and morphological abnormalities	-
*TBPL2*	14q22.3		TATA-Box Binding Protein Like 2	343	Defects in oocyte maturation Female infertility	Alterations Development of the nervous system
*CDC20*	1p34.2	11	Protein Cell Division Cycle 20	499	Oocyte and embryonic development abnormalities	-
*CDC23*	5q31.2		Protein Cell Division Cycle 23	597	Transition from metaphase to anaphase	-
*NLRP5*	19q13.43		NLR Family Pyrin Domain Containing 5	1200	Female infertility	-
*LHX8*	11p31.1		LIM Homeobox 8	356	Egg maturation arrest Ovarian, primary failure	Odontoma and geotrichosis
*WEE1 2* *RAE2*	7q34	12	WEE2 Oocyte Meiosis Inhibiting Kinase	567	Absence of fertilization and zygotes	-
*REC114*	15q24	6	REC114 Meiotic Recombination Protein	266	Arrest of embryonic maturation and development	Distal Arthrogryposis Type 1c
*RAC1*	7p22.1		Rac Family Small GTPase 1	192	Altered mouse oocyte maturation	Serious CNS problems
*PABPC1L*	20q13.12		Poly(A) Binding Protein Cytoplasmic 1 Like	614	Female infertility related to problems in the zona pellucida Maturation arrest	-

## Data Availability

This study analyzed publicly available datasets. The data can be found here: https://pubmed.ncbi.nlm.nih.gov, accessed on 1 August 2024.

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
