# Peer review of "Genetic Abnormalities of Oocyte Maturation: Mechanisms and Clinical Implications"

_ijms, 2024, doi:10.3390/ijms252313002_

Round 1

Reviewer 1 Report

Comments and Suggestions for Authors

In the present work, Baldini et al. try to review the genetic abnormalities of oocyte maturation in mechanisms and clinical implications. Genetic abnormalities in oocyte maturation present significant fertility and embryonic development challenges. The intricate mechanisms of nuclear and cytoplasmic maturation, emphasizing the genetic and molecular factors contributing to oocyte quality and competence are reviewed in this manuscript. Suggesting that identifying specific genetic mutations and their clinical implications offers potential pathways for enhancing fertility treatments and reproductive outcomes. However, there are some questions that should be explained.

Major concerns

1. The genetic abnormalities of oocyte maturation may be avoided by genetic selection. What reasons could lead to genetic abnormalities of oocyte maturation, these should be included.

2. There a high similarity rate of 50%, which is not permissible for a review article. The similarity rate should be decreased.

3. As a review, there is no a fine figure to explain the causes and mechanisms of the genetic abnormalities of oocyte maturation. Some fine figures should be added.

4. English grammar and writing style should be checked and revised throughout the manuscript. I suggest that it is supported by a professional English language proofreading service. There are many paragraphs without a reference, and there is only a reference for almost all paragraphs.

Minor concerns

1. A conclusive sentence should be added in the end of Abstract section.

2. Lines 39-57, there is no a reference.

3. Line 47, ‘in vitro fertilization(IVF)’ should be revised.

4. The aim of this manuscript should be added in the end of the Introduction section.

5. Lines 72-89, revise to a paragraph.

6. Line 96, Chromosomal segregation errors, numbered as 2.1.1, which should be checked throughout the manuscript.

7. Line 141, change ‘Cui L. et al.’ to ‘Cui et al.’, which should be checked throughout the manuscript.

8. Lines 165-168, add a reference.

9. Lines 220-234, add references.

10. Lines 360-364, add a reference.

11. Table 1, references should be added.

12. Lines 426-438, add references.

13. Lines 434-444, add references.

14. Lines 508-545, there is no a reference.

15. Lines 587-596, add a reference.

16. Line 615, ‘development (33).’

17. The format of all references is not suitable for this Journal.

Comments on the Quality of English Language

The English could be improved to more clearly express the research.

Author Response

In response to the 1°st reviewer: majors concerns

Please find below the answers for the major concerns:

1)The genetic abnormalities of oocyte maturation may be avoided by genetic selection. What reasons could lead to genetic abnormalities of oocyte maturation, these should be included.

  1. Normally, no patient has undergone genetic testing for gene abnormalities related to oocyte maturation, as it would be too complex.
  2. This study aims to understand that certain oocyte or embryonic abnormalities that may occur in assisted reproduction treatments could be linked to a genetic issue.
  3. In fact, as the first reviewer points out, it is not possible to avoid genetic anomalies through genetic selection, unless the patient, aware of a possible genetic anomaly, chooses to undergo an egg donation treatment.
  4. Therefore, while we appreciate the suggestions made by the first reviewer, it is frankly impossible to identify the reasons that could lead to genetic anomalies in oocyte maturation.

2) There a high similarity rate of 50%, which is not permissible for a review article. The similarity rate should be decreased.

We have removed the 50% similarity rate by rewriting the text. The new file has been submitted to plagiarism detection software, which returned a much lower percentage than before.

3) As a review, there is no a fine figure to explain the causes and mechanisms of the genetic abnormalities of oocyte maturation. Some fine figures should be added.

 We have added an explanatory figure regarding the genes involved in oocyte maturation.

4) English grammar and writing style should be checked and revised throughout the manuscript. I suggest

that it is supported by a professional English language proofreading service. There are many paragraphs

without a reference, and there is only a reference for almost all paragraphs.

  1. a) The manuscript has been rewritten in English and reviewed by a native speaker.
  2. b) We have added references where they were missing and included additional citations in all paragraphs.

In response to the 1°st reviewer : Minor concerns

1) 1.A conclusive sentence should be added in the end of Abstract section.

A conclusive sentence was added at the end of the abstract section

2) Lines 39-57, there is no a reference.

The Lines were  corrected and  a new reference was added.

  1. Line 47, ‘in vitro fertilization(IVF)’ should be revised.

The sentence was amended

  1. The aim of this manuscript should be added in the end of the Introduction section

The aim of the manuscript was added at the end of the introduction

  1. Lines 72-89, revise to a paragraph.

It was done

  1. Line 96, Chromosomal segregation errors, numbered as 2.1.1, which should be checked throughout the manuscript.

It was done

  1. Line 141, change ‘Cui L. et al.’ to ‘Cui et al.’, which should be checked throughout the manuscript.

It has been corrected

  1. Lines 165-168, add a reference.

done

  1. Lines 220-234, add references.

done

  1. Lines 360-364, add a reference.

Done

  1. Table 1, references should be added.

It was added

  1. Lines 426-438, add references.

Done

  1. Lines 434-444, add references.

done

  1. Lines 508-545, there is no a reference.

done

  1. Lines 587-596, add a reference.

Done

  1. Line 615, ‘development (33).’

done

  1. The format of all references is not suitable for this Journal.

done

Reviewer 2 Report

Comments and Suggestions for Authors

This is a review article based on a database search. While it is acceptable, it lacks the input and commentary of an experienced embryologist or geneticist. Before considering it for publication, please address the following:

Please change ripening medium to maturation medium throughout the text and in general try to better link paragraphs throughout the text

Lines 39-40 – please divide the sentence, the information on the non-optimal quality place with the oocytes and the fertilisation failures develop in 1-2 sentences.

Line 133-139: please move to

Line 196: please provide citation for the oocyte arrest and myocardial infarction

Line 217: to stop what? What is the connection between the ectopic pregnancy and oocyte ability, in ectopic is it better? Also please provide citation here

Line 250: he or they?

Line 361: polar body (please correct throughout the text)

Line 379: germinal (please correct throughout the text)

Line 384: what molecule?

Line 420: provide citation

Table 1: please provide column with relevant citations for each row

Line 464: what stop stands for?

Line 511: please reformulate: what are the age-related changes?

Lines 638-642: provide citation

Line 673: please explain why ‘isthmocele’ was the key word

Please state the whether the AI engines were involved during writing the manuscript

Author Response

In response to the 2° reviewer

First and foremost, we would like to sincerely thank the reviewer for all the corrections and valuable suggestions provided to improve the manuscript.

Please find below the answers for the concerns:

1) Lines 39-40 – please divide the sentence, the information on the non-optimal quality place with the oocytes and the fertilisation failures develop in 1-2 sentences.

The lines were removed from the manuscript

2) Line 133-139: please move to ????

Where and what  we should move? We genuinely did not understand the request.

3) Line 196: please provide citation for the oocyte arrest and myocardial infarction

The citation was provided , however that sentence is not referred to the myocardial infarction but a the oocyte maturation arrest. It was just a mistake in translation.

4) Line 217: to stop what? What is the connection between the ectopic pregnancy and oocyte ability, in ectopic is it better? Also please provide citation here

Although the TUBB8 mutation causes oocyte abnormalities or embryo arrest, in some cases with this anomaly, there have been ectopic or biochemical pregnancies, but never any that were viable.

5)Line 250: he or they?

We changed the sentence

6) Line 361: polar body (please correct throughout the text)

The paragraph  was amended

7)Line 379: germinal (please correct throughout the text)

done

8) Line 384: what molecule?

done

9) Line 420: provide citation

Citation was provided

Table 1: please provide column with relevant citations for each row

done

Line 464: what stop stands for?

The sentence was changed. ‘’Stop’’ was referred to oocyte maturation arrest

Line 511: please reformulate: what are the age-related changes?

The sentence was reformulated

Lines 638-642: provide citation

The citation was provided

Line 673: please explain why ‘isthmocele’ was the key word

It was a mistake and the sentence was amended.

Please state the whether the AI engines were involved during writing the manuscript

none

Reviewer 3 Report

Comments and Suggestions for Authors

Overall Recommendation

 The manuscript is an interesting review of current knowledge on the genetic factors of oocytes maturation. The authors have presented a lot of clinical aspects of pointed subjects what makes the paper applicable.

The title:

Genetic abnormalities of oocyte maturation: mechanisms and 2 clinical implications

According to the reviewer, the title accurately reflects the content of the manuscript.

 The structure of the review work is very clear and allows for a clear reception of the difficult topic collected and finding an interesting fragment for the citing author.

The cited references are in 72 % recent publications (within the last 10 years) in which 90% are from the last 4 years, which proves that the references are well prepared, but also that the chosen topic is relevant. Manuscript doesn’t include self-citations.

  The manuscript presents current knowledge, where no similar article was published recent years. The subject is discussed quite generally - however, in reviewer opinion this is dictated by the broadness of the issue and the clinical approach to the review.

Author Response

REVIEWER 3

First and foremost, we would like to sincerely thank the reviewer for all the corrections and valuable suggestions provided to improve the manuscript.

“The manuscript is an interesting review of current knowledge on the genetic factors of oocytes maturation. The authors have presented a lot of clinical aspects of pointed subjects what makes the paper applicable.

The title:

Genetic abnormalities of oocyte maturation: mechanisms and 2 clinical implications

According to the reviewer, the title accurately reflects the content of the manuscript.

 The structure of the review work is very clear and allows for a clear reception of the difficult topic collected and finding an interesting fragment for the citing author.

The cited references are in 72 % recent publications (within the last 10 years) in which 90% are from the last 4 years, which proves that the references are well prepared, but also that the chosen topic is relevant. Manuscript doesn’t include self-citations.

 The manuscript presents current knowledge, where no similar article was published recent years. The subject is discussed quite generally - however, in reviewer opinion this is dictated by the broadness of the issue and the clinical approach to the review.”

We thank you for fully understanding the reason for writing this article and the possible impact on the clinical activity of assisted reproduction techniques. We are also grateful to the reviewer for the judgment expressed on our paper.

Round 2

Reviewer 1 Report

Comments and Suggestions for Authors

Thanks for author’s responses. However, there are many questions that should be explained.

1. As a Review article, recent related articles should be referred. However, lots of recent related articles have not been referred. For example,

Liu C, Zuo W, Yan G, Wang S, Sun S, Li S, Tang X, Li Y, Cai C, Wang H, Liu W, Fang J, Zhang Y, Zhou J, Zhen X, Feng T, Hu Y, Wang Z, Li C, Bian Q, Sun H, Ding L. Granulosa cell mevalonate pathway abnormalities contribute to oocyte meiotic defects and aneuploidy. Nat Aging. 2023;3(6):670-687.

Zhao B, Li H, Zhang H, Lan X, Ren X, Zhang L, Ma H, Zhang Y, Wang Y. Inhibition of HSP90AA1 induces abnormalities in bovine oocyte maturation and embryonic development. Reproduction. 2024;167(5):e230411.

Wang X, Zhou R, Lu X, Dai S, Liu M, Jiang C, Yang Y, Shen Y, Wang Y, Liu H. Identification of nonfunctional PABPC1L causing oocyte maturation abnormalities and early embryonic arrest in female primary infertility. Clin Genet. 2023;104(6):648-658.

Ebru H, Dahan MH, Sezer O, BaÅŸbuÄŸ A, Kaan H, Güngör ND, Baltacı V, Tan SL, Åžafak H. TUBB8 mutations as a cause of oocyte maturation abnormalities: presentation of oocyte and embryo profiles and novel mutations. Reprod Biomed Online. 2023;47(5):103257.

2. This review point out that English grammar and writing style should be checked and revised throughout the manuscript. However, there are too many low wrongs in the revised manuscript. The writing level of authors is lower. For example,

Line 26, ‘(IVF)’

Introduction section, there are too many short paragraphs.

Throughout this manuscript, there are so many ‘metaphase II (MII)’, ‘germinal vesicle (GV)’, and others.

Lines 87-102 should be deleted.

Line 184, ‘Loeuillet, C. et al.’, there are many such wrong throughout this manuscript.

3. This review point out that the format of all references is not suitable for this Journal. However, the format of all references is still not suitable for this Journal.

Comments on the Quality of English Language

The English could be improved to more clearly express the research.

Author Response

In responding to the first reviewer round 2

  1. As a Review article, recent related articles should be referred. However, lots of recent related articles have not been referred.

Liu C, Zuo W, Yan G, Wang S, Sun S, Li S, Tang X, Li Y, Cai C, Wang H, Liu W, Fang J, Zhang Y, Zhou J, Zhen X, Feng T, Hu Y, Wang Z, Li C, Bian Q, Sun H, Ding L. Granulosa cell mevalonate pathway abnormalities contribute to oocyte meiotic defects and aneuploidy. Nat Aging. 2023;3(6):670-687.

( IT WAS ADDED)

Wang X, Zhou R, Lu X, Dai S, Liu M, Jiang C, Yang Y, Shen Y, Wang Y, Liu H. Identification of nonfunctional PABPC1L causing oocyte maturation abnormalities and early embryonic arrest in female primary infertility. Clin Genet. 2023;104(6):648-658.

( IT WAS ADDED)

Ebru H, Dahan MH, Sezer O, BaÅŸbuÄŸ A, Kaan H, Güngör ND, Baltacı V, Tan SL, Åžafak H. TUBB8 mutations as a cause of oocyte maturation abnormalities: presentation of oocyte and embryo profiles and novel mutations. Reprod Biomed Online. 2023;47(5):103257.

( IT WAS ADDED)

Zhao B, Li H, Zhang H, Lan X, Ren X, Zhang L, Ma H, Zhang Y, Wang Y. Inhibition of HSP90AA1 induces abnormalities in bovine oocyte maturation and embryonic development. Reproduction. 2024;167(5):e230411.

(IT WAS NOT ADDED): In drafting the article, we intentionally chose not to include bibliographic references related to animal models, as a methodological choice. The only exception is the bibliographic reference concerning oocytes of a certain type, where a translational mechanism is described that may also be applicable to humans. This reference was included because the translational process discussed holds relevance that could extend to human studies, making it pertinent to our research context as well.

  1. This review point out that English grammar and writing style should be checked and revised throughout the manuscript. However, there are too many low wrongs in the revised manuscript. The writing level of authors is lower. For example,

Line 26, ‘(IVF)’  (it was amended)

Lines 87-102 should be deleted.   (it was deleted)

Line 184, ‘Loeuillet, C. et al.’, there are many such wrong throughout this manuscript. (corrected)

Thank you for the feedback. However, we would like to remind you that our native speaker reviewed and edited the text, suggesting that we use smaller paragraphs to make the text clearer and more fluid.

3) Throughout this manuscript, there are so many ‘metaphase II (MII)’, ‘germinal vesicle (GV)’, and others

Unfortunately, there are no alternative terms to indicate the stages of oocyte development or maturation. There are no synonyms for Germinal Vesicle or Metaphase 2.

4)  This review point out that the format of all references is not suitable for this Journal. However, the format of all references is still not suitable for this Journal.

The reference format has been adjusted to comply with the journal's guidelines.

Round 3

Reviewer 1 Report

Comments and Suggestions for Authors

There is no author's reply for this third review. However, there are still many questions that should be explained. We do not think that this manuscript is suitable for publishing in IJMS at present.

1. As a Review article, there is only a figure, which can not cover all the contents of this manuscript. More high quality figures are needed.

2. This review point out that English grammar and writing style should be checked and revised throughout the manuscript. However, there are too many low wrongs in the revised manuscript. The writing level of authors is lower, which is suitable for publishing in IJMS at present. For example,

Introduction section, there are too many short paragraphs.

‘Cui et al’ changes to ‘Cui et al’, ‘Loeuillet, C. et al.’ changes to ‘Loeuillet, C.’, and ‘Yao, Z. et al. 2022’ changes to ‘Yao, Z. 2022’, et al., which are not right.

3. This review point out that the format of all references is not suitable for this Journal. However, the format of all references is still not suitable for this Journal.

Comments on the Quality of English Language

The English could be improved to more clearly express the research.

Author Response

Dear reviewer

You are right; we realized we have not submitted the latest version of the corrected file.

  1. As a review article, there is only one figure, which cannot cover the entire content of this manuscript. Higher quality data is needed.We added two more figures to implement the text explanation, as you suggested.
  2. This review emphasizes that the English grammar and writing style should be checked and revised throughout the manuscript. However, there are too many low errors in the revised manuscript. The writing level of the authors is lower, which is suitable for publication in IJMS at present. For example Introductory section, there are too many short paragraphs. We have revised the entire paper again, corrected the English errors, and merged many short paragraphs into groups that are more suitable for publication. All abbreviations have been corrected.
  3. This review emphasizes that the format of all references is not suitable for this journal. However, it is not suitable for this journal yet.

The references have been reviewed and corrected

Thanks again for your availability

Round 4

Reviewer 1 Report

Comments and Suggestions for Authors

Thanks for author’s responses. However, English grammar and writing style still should be checked and revised throughout the manuscript. For example,

Line 404, changes ‘Wang W.’ to ‘Wang et al.’.

Line 474, changes ‘Wyse B.A.’ to ‘Wyse et al.’. Please check this throughout the manuscript.

In addition, Figures 2 and 3, figures legends should be added.

Comments on the Quality of English Language

The English could be improved to more clearly express the research.

Author Response

We have taken steps to accept all revisions and complete the captions of the figures.

Best regard